# DSC-SeNet: Unilateral Network with Feature Enhancement and Aggregation for Real-Time Segmentation of Carbon Trace in the Oil-Immersed Transformer

**DOI:** 10.3390/s25010043

**Published:** 2024-12-25

**Authors:** Liqing Liu, Hongxin Ji, Junji Feng, Xinghua Liu, Chi Zhang, Chun He

**Affiliations:** 1State Grid Tianjin Electric Power Research Institute, Tianjin 300180, China; liulq328@126.com (L.L.); galago_@hotmail.com (J.F.); zc_zhangchi@163.com (C.Z.); chun.he@tj.sgcc.com.cn (C.H.); 2School of Electrical Engineering, China University of Mining and Technology, Xuzhou 221116, China; lxh9357@163.com

**Keywords:** oil-immersed transformer, discharge carbon trace, semantic segmentation, deformable convolution, canny edge detection, real-time processing

## Abstract

Large oil-immersed transformers have metal-enclosed shells, making it difficult to visually inspect the internal insulation condition. Visual inspection of internal defects is carried out using a self-developed micro-robot in this work. Carbon trace is the main visual characteristic of internal insulation defects. The characteristics of carbon traces, such as multiple sizes, diverse morphologies, and irregular edges, pose severe challenges for segmentation accuracy and inference speed. In this paper, a feasible real-time network (deformable-spatial-Canny segmentation network, DSC-SeNet) was designed for carbon trace segmentation. To improve inference speed, a lightweight unilateral feature extraction framework is constructed based on a shallow feature sharing mechanism, which is designed to provide feature input for both semantic path and spatial path. Meanwhile, the segmentation model is improved in two aspects for better segmentation accuracy. For one aspect, to better perceive diverse morphology and edge features of carbon trace, three measures, including deformable convolution (DFC), Canny edge operator, and spatial feature refinement module (SFRM), were adopted for feature perception, enhancement, and aggregation, respectively. For the other aspect, to improve the fusion of semantic features and spatial features, coordinate attention feature aggregation (CAFA) is designed to reduce feature aggregation loss. Experimental results showed that the proposed DSC-SeNet outperformed state-of-the-art models with a good balance between segmentation accuracy and inference speed. For a 512 × 512 input, it achieved 84.7% mIoU, which is 6.4 percentage points higher than that of the baseline short-term dense convolution network (STDC), with a speed of 94.3 FPS on an NVIDIA GTX 2050Ti. This study provides technical support for real-time segmentation of carbon traces and transformer insulation assessment.

## 1. Introduction

Large oil-immersed transformers are the primary type of power conversion equipment in modern power systems. Regular condition monitoring is essential to prevent potential issues. However, the metal-enclosed shell of oil-immersed transformers makes it impossible to directly observe the internal conditions. Indirect monitoring techniques, such as dissolved gas analysis, the three-ratio method [1,2], and the improved three-ratio method [3,4], often fail to accurately identify the type, location, and severity of internal faults [5,6,7,8]. As a direct maintenance method, visual inspection is widely used due to the advantages of quickly locating and intuitively assessing faults. However, common visual inspection methods have drawbacks of long maintenance cycles, high risk of injury to personnel and equipment, and high costs associated with power outages for repairs.

With advances in robotics and artificial intelligence, internal inspection micro-robots have become a highly effective solution to replace traditional methods. These compact robots have excellent maneuverability and high inspection efficiency. They can autonomously enter the transformers through hand holes, quickly and intuitively identifying internal defects along with their locations, types, and severities. In 2018, ABB launched Txplore, an oil-immersed transformer inspection robot, which showcased innovative design concepts and technologies [9,10,11]. Researchers from Shenyang Ligong University developed the SSTIR, a spherical inspection robot inspired by underwater robotic structures [12,13,14]. This spherical robot could navigate inside oil-filled transformer tanks using an oil jet propulsion system and perform visual inspections. Additionally, Tsinghua University, China University of Mining and Technology, and Tianjin Electric Power Research Institute have jointly developed an intelligent transformer inspection robot, focusing on key areas such as oil-based positioning, defect recognition, and inspection path planning. This collaborative work has advanced the capabilities of transformer internal inspection technologies [15,16,17,18,19].

Visual recognition and analysis of defect targets are critical technical components for the micro-robot to autonomously perform inspection. Discharge carbon traces are key visual indicators of internal insulation faults. And its accurate semantic segmentation is essential for evaluating discharge intensity and assessing insulation degradation trends. However, research on the segmentation of carbon traces is limited. Both segmentation accuracy and inference speed require significant improvements to achieve precise and real-time fault detection inside the transformer.

As one of the core tasks of computer vision, semantic segmentation assigns a semantic label to each pixel, allowing pixels with the same label to share common visual attributes. It is widely used in areas such as augmented reality, autonomous driving, scene understanding, and video surveillance. With the advancement of convolutional neural networks, frameworks such as UNet [20] and DeepLab [21] have greatly improved segmentation accuracy. The UNet framework uses an encoder–decoder structure with skip connections, demonstrating effective applications in medical diagnostics. To further improve the segmentation performance, several UNet-based networks have been designed, including R2U-Net [22], DCSAU-Net [23], nnU-Net [24], and Res-UNet [25]. However, the high computational complexity of high-resolution features in the U-shaped encoding–decoding structure reduces the inference speed. To address this problem, new CNN-based segmentation models have been developed to balance accuracy and efficiency. For example, ICNet reduces model complexity by adjusting input resolution, providing a simple but effective solution [26]. However, it suffers from a loss of spatial detail at object boundaries. ENet, on the other hand, employs pruning techniques to reduce network channels and depth, creating a more compact structure with faster inference speed, but at the cost of a smaller receptive field [27]. To overcome the drawback of detailed features easily overwhelmed by surrounding contextual information, PIDNet incorporated a proportional integral (PI) controller into the two-branch network, which contributed to the segmentation accuracy [28]. These models typically sacrifice spatial detail to achieve faster inference, compromising segmentation accuracy. To address these trade-offs, SFNet introduced the flow alignment module to enhance feature alignment and fusion within each layer [29]. To overcome the speed-accuracy limitations of encoder–decoder structures, Yu et al. proposed the bilateral segmentation network (BiSeNet), a pioneering dual-path network with a spatial path and a context path [30]. However, since the context path of BiSeNet is designed according to classification networks, it lacks full adaptability to segmentation tasks. To address this, the short-term dense concatenation network (STDC) introduces an STDC module, which improves the perception ability for multi-scale information [31]. Furthermore, using the STDC network as the backbone, BiSeNetV3 removes the spatial path while adding feature refinement and fusion modules. It enhances segmentation performance while accelerating inference speed [32]. Benefitting from the broad global receptive field, transformer structure provides a new path to tackle semantic segmentation challenges. SCTNet utilizes a transformer as the training-only semantic branch, considering its superb ability to extract long-range contexts [33]. HAFormer combines the hierarchical feature extraction ability of CNNs and the efficient transformer (ET) module to tackle lightweight semantic segmentation challenges [34]. Liu et al. carried out meaningful segmentation research in clinical diagnosis. For glioma segmentation, they developed a multi-modal magnetic resonance (MR) image fusion method based on the adversarial learning framework [35]. For brain tumor segmentation, they respectively adopted the Swin transformer and Sobel operator for semantic feature extraction and edge feature enhancement, providing good insights for our research [36].

As the key visual indicator of insulation faults, discharge carbon trace is characterized in multiple scales, various morphology, and edge irregularity, posing severe challenges for both segmentation accuracy and inference speed. To address this, we develop a lightweight semantic segmentation network specially for carbon trace segmentation. In this paper, a unilateral feature extraction framework with feature enhancement and aggregation improvements was proposed to strike a better balance between segmentation accuracy and inference speed.

The main contributions of this paper are three-fold:(1)To reduce computational requirements, a lightweight unilateral feature extraction framework is constructed with the STDC module as the backbone. It could provide feature input for both semantic path and spatial path based on a shallow feature sharing mechanism.(2)To better extract morphological and edge features of carbon traces, DFC module, Canny edge operator, and SFRM module are sequentially designed and integrated into the unilateral framework for feature perception, enhancement, and aggregation, respectively.(3)To enhance feature aggregation of deep semantic features and shallow spatial features, the CAFA is designed based on a coordinate attention mechanism. It could reduce feature fusion loss through feature cross-fusion and alignment in multiple dimensions.

## 2. Related Work

### 2.1. Brief Introduction of Inspection Micro-Robot

Building on robotics and artificial intelligence technologies, an internal inspection micro-robot for oil-immersed transformers was developed in our previous work. The overall structure of the robot, shown in Figure 1, is composed of a shell body, ultrasonic emission sensor, image acquisition module, ultrasonic ranging module, vertical and horizontal propeller propulsion devices, pressure sensor, and remote-control system [15,16].

As the inspection micro-robot navigates through the transformer oil, the image acquisition module captures images of various internal components and areas. The inspection micro-robot was designed with the Jetson Xavier NX 16 GB. The installation of the robot operating system (ROS) took up 11.7 GB, leaving only 4.3 GB of storage. Therefore, a 32 GB SD card is used as external storage for carbon trace image storage needs. When the internal inspection is completed with the micro-robot, the image data could be transmitted to the external remote-control platform, allowing either manual inspection of insulation defects or automated feature extraction and identification using computer vision algorithms.

### 2.2. Previous Research on Discharge Carbon Trace Recognition and Segmentation

To support the fully autonomous operation of the inspection micro-robot, we previously developed a carbon trace recognition model based on an improved YOLOv8 algorithm [17]. This model enables rapid and accurate identification of carbon traces from the large sets of transformer interior images collected by the inspection micro-robot, as shown in Figure 2, achieving a recognition accuracy of 91.6% and a processing speed of 99.2 FPS. Building on the success of carbon trace recognition, it is necessary to accurately segment carbon trace images for precise assessment of discharge types and severity. To this end, an effective carbon trace segmentation model was designed, named HSP-UNet [19], which achieved promising results in the segmentation of discharge carbon traces, as shown in Figure 3. However, due to the limitations of the encoder–decoder structure, further improvements in segmentation accuracy and inference speed are needed.

## 3. Proposed Method

### 3.1. Overall Structure of DSC-SeNet

Discharge carbon traces are characterized by diverse morphologies, large-scale variations, and irregular edges, which pose significant challenges for both segmentation accuracy and inference speed. To this end, a novel semantic segmentation model, DSC-SeNet, is proposed in this paper, which tries to strike a balance between segmentation precision and inference speed. Inspired by BiSeNetV3, DSC-SeNet adopts a unilateral network for feature extraction by using a shallow feature-sharing mechanism. This mechanism allows shallow features to serve dual roles as both semantic and spatial features, thereby reducing model complexity and improving inference speed. In other words, the bilateral framework respectively uses a semantic path and a spatial path for semantic and spatial feature extraction. Unlike the bilateral framework, as shown in Figure 4, the unilateral framework modifies the backbone by removing the spatial path and using the single-path STDC network for shallow feature extraction. To address the insufficient perception of carbon trace spatial detail, the DFC is integrated into the STDC module. To improve the perception and fusion of multi-scale spatial features, the SFRM is designed to optimize the weighting parameters of multi-scale features. In addition, to improve the segmentation accuracy of carbon trace edges, an edge feature enhancement branch based on the Canny operator is introduced. Finally, to strengthen the fusion of deep semantic features with shallow spatial features, the CAFA module is designed based on the coordinate attention mechanism. It is incorporated as the final layer in the feature extraction network, which minimizes feature loss during the fusion process. The overall structure of the DSC-SeNet is shown in Figure 4.

### 3.2. Proposed STDDC

To improve the adaptability for semantic segmentation tasks, STDC is used as the core component in the feature extraction backbone. In image classification tasks, a common approach is to increase the number of channels in deeper layers to improve classification accuracy by exploiting multi-channel image features. In semantic segmentation, however, the focus shifts to achieving a scalable receptive field and capturing multi-scale information. Shallow layers require enough channels to encode fine-grained details, while deeper layers benefit from an expanded receptive field that emphasizes semantic information. To avoid redundancy in the multi-channel structure of a shallow network while preserving feature richness in deeper layers, STDC uses skip connections to concatenate feature maps from *x*_1_ to *x*_n_, forming the output of the feature extraction module, as shown in Figure 5a. The final output of the STDC module is defined by Equation (1).
(1)$$x_{\mathrm{output}}=F(x_{1},x_{2},\cdots ,x_{n})$$
where *x*_output_ denotes the final output of the STDC module, while *F* represents the feature fusion function. To achieve an optimal balance between computational efficiency and fusion performance, feature concatenation is employed as the fusion function. The terms $x_{1},x_{2},\cdots ,x_{n}$ refer to the feature maps of different convolutional layers. Considering the multi-scale characteristics of carbon traces, the STDC module adopts a four-layer convolutional design.

While the STDC module effectively expands the receptive field of deeper layers and reduces channel redundancy in shallower layers, it uses standard convolution modules with fixed-size (e.g., 3 × 3, 5 × 5, and 7 × 7) rectangular receptive fields to extract features from input images. These fixed sampling points (e.g., 9 points for a 3 × 3 kernel) are evenly distributed on a regular grid, assuming that the target features are aligned with the rectangular shape of the receptive field. However, carbon trace images often exhibit varying scales and irregular morphologies. In addition, this approach assumes a fixed proportional relationship between the receptive field and the target feature, but carbon traces vary widely in feature size. Consequently, standard convolution faces challenges in capturing the irregular and variable-scale characteristics of carbon traces. To address these limitations, the DFC is introduced in the STDC module, which exploits the spatial flexibility of deformable receptive fields to more effectively adapt to the morphological diversity of carbon traces, as shown in Figure 5b. DFC is an innovative feature extraction technique that overcomes the limitations of standard convolutions by dynamically adjusting the receptive field [37]. By applying an independent offset field, DFC adjusts the positions of sampling points within the receptive field according to the shape of the target, enabling the receptive field to modify its shape, contour, and even orientation. This approach enhances the ability to capture multi-scale and diverse morphological features of carbon traces, as depicted in Figure 6. For example, if both input and output feature maps are 4 × 4 × 1, the network structure of a 3 × 3 deformable convolution is illustrated in Figure 7.

### 3.3. Proposed SFRM

The BiSeNet network incorporates an attention refinement module to improve feature extraction within the spatial path [30]. However, this module utilizes global average pooling to perceive target features, producing a feature map with the dimensions of C × 1 × 1. This approach quantifies the importance of each channel based only on inter-channel relationships and overlooks the critical role of spatial information in capturing target features. In practice, spatial relationships for neighboring pixels are essential for achieving high semantic segmentation accuracy, and the spatial details of an image significantly influence the formation of the attention map. Inspired by the coordinate attention mechanism [38], the SFRM is designed to enhance the spatial information of target features. The core idea of the coordinate attention mechanism is to perform operations such as pooling and convolution along the height (H) and width (W) dimensions individually, allowing more precise localization and identification of features of interest. It improves the ability to capture detailed edge features of carbon traces. First, the input feature map is averaged along the height (H) and width (W) dimensions, yielding two directional feature maps, as shown in Equation (2). These maps are then concatenated, followed by Conv2d, BatchNorm, and sigmoid operations, resulting in a feature map of dimension 1 × (W + H) × C/r), as shown in Equation (3). This map is then split along the channel dimension into tensors $f^{h}$ and $f^{w}$, which are processed using Conv2d and sigmoid functions to generate attention weights $g^{h}$ and $g^{w}$, as shown in Equation (4). Finally, these weights are applied to the input feature map $x_{c}\left(i,j\right)$, yielding the output feature map $y_{c}\left(i,j\right)$, as shown in Equation (5).
(2)$$\left\{\begin{array}{l} Z_{c}^{h}\left(h\right)=\frac{1}{W}\sum_{0\leq i<W}\left|x_{c}\left(h,i\right)\right.\\ Z_{c}^{w}\left(w\right)=\frac{1}{H}\sum_{0\leq j<H}\left|x_{c}\left(j,w\right)\right. \end{array}\right.$$


(3)
$$f=\delta \left(F_{1}\left(\left[z^{h},z^{w}\right]\right)\right)$$



(4)
$$\left\{\begin{array}{l} g^{h}=\delta \left(F_{h}\left(f^{h}\right)\right)\\ g^{w}=\delta \left(F_{w}\left(f^{w}\right)\right) \end{array}\right.$$



(5)
$$y_{c}\left(i,j\right)=x_{c}\left(i,j\right)\times g_{c}^{h}\left(i\right)\times g_{c}^{w}\left(j\right)$$


Based on the concept of a coordinate attention mechanism, the proposed SFRM module processes the spatial path input feature map and the Canny edge feature map by splitting them along the height (H) and width (W) dimensions, yielding feature maps of (C × H × 1) and (C × 1 × W), respectively. These feature maps are then concatenated and fused along the H and W directions, followed by convolution, normalization, and other processing steps. This approach allows the feature map of each channel to encode carbon trace features along the horizontal and vertical directions at the spatial scale, facilitating a more coherent integration of the two spatial features. This, in turn, improves the extraction and emphasis of carbon trace edge features. Finally, a sigmoid operation produces an attention vector, which is combined with the two original input feature maps through vector addition to produce a refined and enhanced spatial feature map. This output improves the ability to capture the spatial detail of carbon trace. The network structure of the SFRM module is shown in Figure 8.

### 3.4. Edge Feature Enhancement

The complex edges of discharge carbon traces pose a significant challenge to segmentation accuracy. Traditional edge detection operators, which use gradient information to capture image edges, provide a valuable complement to the spatial features of carbon traces extracted by the STDDC module. Therefore, Canny operator is used to generate edge features that refine and enhance the areas where edges appear blurred [39]. Compared to other edge detection methods, Canny operator offers advantages such as resistance to noise and strong detection capability for weak edges, making it well suited for extracting the complex, multi-scale edge features of carbon traces. Canny edge detection is a dual-threshold gradient information-based algorithm with the following steps: First, the initial image is filtered by a two-dimensional Gaussian function, as specified in Equation (6), yielding the smoothed output shown in Equation (7). Next, the first derivatives in the X and Y directions are calculated according to Equation (8), denoted as $K_{x}(x,y)$ and $K_{y}(x,y)$, respectively. These derivatives are then used to compute the gradient magnitude M(x,y) and gradient direction θ(x,y), as given in Equations (9) and (10). Non-maximum suppression is then applied to further refine the edges, and finally edge detection and connectivity are achieved according to the presented high and low thresholds.
(6)$$G(x,y)=\frac{1}{2\pi \sigma^{2}}\exp\left(-\frac{x^{2}+y^{2}}{2\sigma^{2}}\right)$$


(7)
I(x,y)=G(x,y)×F(x,y)



(8)
$$\left\{\begin{array}{l} K_{x}(x,y)=\frac{I(i,j+1)-I(i,j)+I(i+1,j+1)-I(i+1,j)}{2}\\ K_{y}(x,y)=\frac{I(i,j)-I(i+1,j)+I(i,j+1)-I(i+1,j+1)}{2} \end{array}\right.$$



(9)
$$M(x,y)=\sqrt{{K^{2}}_{x}(x,y)+{K^{2}}_{y}(x,y)}$$



(10)
$$\theta (x,y)=\arctan\left(\frac{K_{x}(x,y)}{K_{y}(x,y)}\right)$$


Since the Canny operator uses gradient vectors as the basis for edge detection, this study applies the Canny algorithm at 1/8 and 1/16 scales of the input image to reduce computational cost and maintain inference speed. Through multiple trials, the high and low thresholds of the Canny operator are set at 40 and 120, respectively. With gradient variations and dual-threshold criteria, the Canny algorithm ultimately outputs a binary mask image. The resulting Canny edge detection map is then multiplied by a weighting coefficient determined by the segmentation head, producing an additional spatial feature *E*, as shown in Equation (11). This supplementary feature, combined with the spatial features extracted by the STDDC, is fed into the SFRM module to improve the representation of weak edges and multi-scale features of carbon traces. By incorporating Canny edge features, this method enriches the spatial characteristics while maintaining low computational costs, thereby improving the sensitivity to edge features and increasing the segmentation accuracy for carbon traces.
(11)$$E=\sum_{j=1}^{H}\sum_{i=1}^{W}E_{ij}\times \alpha$$

### 3.5. Proposed CAFA

A key goal in the development of semantic segmentation models has been to preserve both deep semantic information and fine spatial details within the encoded features. Features extracted from the semantic pathway provide rich and deep contextual information, while those from the spatial pathway preserve essential shallow details. However, direct fusion of these two types of features is challenging. To address this, the CAFA module is introduced to effectively integrate semantic and spatial features, as shown in Figure 9.

The process begins by separating the input semantic and spatial features into two branches, each of which undergoes convolution and batch normalization to align the feature map dimensions. Next, the semantic and spatial feature maps are interleaved, resulting in a fused feature map of size (*C*_1_ + *C*_2_) × H/32 × W/32. To address the challenge of edge segmentation caused by the limited spatial detail in carbon trace features, the coordinate attention mechanism [38] is applied. It concatenates feature maps along the height and width dimensions, followed by convolution and normalization steps, encoding two-dimensional spatial information into the carbon trace features and improving segmentation accuracy for weak edges. To further reduce the complexity and computational cost, an SE block is used to compress the number of channels [40]. Finally, the cross-fused carbon trace features from both branches are combined to form an attention-enhanced feature map, which provides a robust basis for pixel-wise classification of carbon trace images.

### 3.6. Booster Training Strategy

To further improve segmentation accuracy, this paper introduces a booster training strategy. Similar to a rocket booster, auxiliary segmentation heads are inserted at different points along the semantic branch during the training process. This approach incorporates segmentation differences from different stages of the semantic branch into the loss function that guides the model training process. During the inference process, these auxiliary segmentation heads are removed to maintain inference speed, as shown in Figure 4. This strategy improves segmentation accuracy while adding minimal computational complexity for the inference. In Section 5.2, the effect of positions and quantity of auxiliary segmentation heads on segmentation performance will be analyzed to optimize the configuration of the booster training strategy.

## 4. Data Collection and Preprocessing of Discharge Carbon Traces

### 4.1. Acquisition of Carbon Trace Images

The internal environment of oil-immersed transformers is characterized by high voltage, high current, sealed metal enclosures, and low light, making the collection of carbon mark samples challenging and high-risk. To address these difficulties, a discharge test platform is constructed to simulate the oil–paper insulation structure inside the oil-immersed transformer, thereby reducing the complexity and risk of sample acquisition. The setup uses a high-resolution industrial camera (HTSUA134GC/M, 1.3 megapixels, 211 FPS, Hua Teng Vision, Shenzhen, China) attached with an extra supplementary illumination source (Ulanzi, 1000 lux@0.5m, Dongguan, China) positioned 25 cm from the oil-immersed paperboard to capture carbon trace samples. The drying level of the paperboard has a significant effect on the morphology of the discharge carbon traces. When the paperboard has a low moisture content and is dry, the carbon traces form dendritic patterns with highly irregular edges, requiring the semantic segmentation model to have a strong detail perception capability. In contrast, when the paperboard has higher moisture content and is damp, the carbon traces become cluster like with relatively regular edges. However, these cluster-like traces feature white speckled areas and black carbonized areas within them, which can lead to boundary misclassification, as shown in Figure 10a,b. In addition, the randomness of surface discharge in the transformer results in carbon traces of different sizes and morphologies, as illustrated in Figure 10c,d, requiring robust global perception and fine-grained feature extraction capabilities of the segmentation model. Considering the unstable lighting conditions of transformer interiors, the illumination intensity was adjusted during image acquisition to capture carbon trace images at different brightness levels, as shown in Figure 10e,f. Differences in contrast and limited clarity in these images increase the challenge of feature extraction for the segmentation model.

### 4.2. Image Enhancement Based on Improved MSRCR Algorithm

In the low-light environment of oil-immersed transformers, challenges such as unstable lighting conditions and glare from oil smudges result in significant variations in brightness and insufficient contrast of carbon trace samples. This makes it difficult to accurately extract carbon trace features. To increase the visibility of these features and improve segmentation accuracy, an improved multi-scale retinex with color restoration (MSRCR) algorithm is used for image enhancement.

Retinex theory models an image as the product of its illumination and reflectance components, with human visual perception being primarily determined by the reflectance component [41]. By applying a Gaussian blur to the original image, the illumination component can be isolated, while the reflectance component, i.e., the enhanced image, can be derived through Equation (12). Extending the retinex theory to multiple scales leads to the MSRCR algorithm, which calculates the reflectance component as described in Equation (13). The algorithm further uses multi-scale Gaussian surround filtering to estimate the original image illumination and applies logarithmic operations on the reflectance component to achieve color restoration and linear stretch enhancement, as expressed in Equation (14). However, the MSRCR algorithm involves numerous empirical parameters that limit its generalization capabilities and introduce defects such as halo effects and color distortion. To handle the complex and intricate edges of carbon traces, the MSRCR algorithm is enhanced with Gaussian and guided filtering functions, as shown in Equations (15) and (16). This improvement achieves superior edge preservation and feature enhancement [38].
(12)$$I\left(x,y\right)=L\left(x,y\right)R\left(x,y\right)$$
where $I\left(x,y\right)$ represents the original image, $L\left(x,y\right)$ denotes the illumination component, and $R\left(x,y\right)$ represents the reflectance component.
(13)$$R_{i}\left(x,y\right)=\sum_{n=1}^{N}W_{n}\left\{\log\left[S_{i}\left(x,y\right)\right]-\log\left[S_{i}\left(x,y\right)\times M_{n}\left(x,y\right)\right]\right\}$$
where $R_{i}\left(x,y\right)$ denotes the reflectance component obtained through the MSRCR algorithm, with the subscript $i\in \left\{r,g,b\right\}$ indicating the three-color channels. *N* represents the number of retinal scales, $W_{n}$ is the weighting factor for each scale, and $M_{n}\left(x,y\right)$ refers to the surround function.
(14)$$R_{MSRCR,i\left(x,y\right)}^{*}=C_{i}\sum_{n=1}^{N}W_{n}R_{n,i\left(x,y\right)}$$
where $R_{n,i}$ represents the reflectance at the *n*th scale for the *i*th color channel, and $C_{i}$ is the color restoration factor used to adjust the proportions of the three-color channels.
(15)$${L^{\prime }}_{n}=F_{guided}\left(S,L_{n},r_{n},\varepsilon \right)$$
where ${L^{\prime }}_{n}$ represents the illumination component, *S* is the original image with guiding features, $L_{n}$ is the Gaussian-filtered image, $r_{n}$ is the guiding filter scale, and ε is the regularization parameter.

By combining Equations (13)–(15), the image reflectance component $R_{i}\left(x,y\right)$ can be rewritten as ${R^{\prime \prime }}_{i}\left(x,y\right)$, which is shown in Equation (16).
(16)$${R^{\prime \prime }}_{i}\left(x,y\right)=\sum_{n=1}^{N}W_{n}\left\{\log\left[S_{i}\left(x,y\right)\right]-\log\left[{L^{\prime }}_{i}\left(x,y\right)\right]\right\}$$

The improved MSRCR algorithm was applied to improve the images of carbon trace samples, as illustrated in Figure 11. Figure 11a,c show the original carbon trace images, which suffer from low brightness and poor contrast, resulting in blurred edges that make it difficult to extract edge features. After applying the improved MSRCR algorithm to Figure 11a,c, the images show a significant increase in brightness and sharpness, with much clearer carbon trace edges. This improvement reduces the difficulty of extracting edge features and improves the accuracy of semantic segmentation, as shown in Figure 11b,d. Therefore, the subsequent study and comparative experiment are conducted on this basis that all the carbon trace images are processed with the improved MSRCR algorithm.

### 4.3. Data Augmentation and Dataset Construction

In the metal-enclosed oil tank of the transformer, collecting carbon trace samples is both challenging and hazardous, which limits the availability of sample data. To improve the generalization and segmentation performance of the model, there are two kinds of data augmentation methods, i.e., processing based and training based [42]. Although having the advantage of increasing the dataset diversity and complexity, the training-based methods suffer from implementation complexity and high computational cost. Because the aim of our research is to construct a lightweight segmentation network with limited training resources, this study employed processing-based methods—including horizontal and vertical flipping, translation, scaling, and Gaussian blur—to expand the original dendritic carbon trace samples to 2495 and the clustered samples to 2825. These augmented samples were then divided into training, validation, and test sets in an 8:1:1 ratio. Specifically, the dendritic carbon trace dataset contains 1996 training samples, 250 validation samples, and 249 test samples, while the clustered dataset includes 2260 training samples, 283 validation samples, and 282 test samples. To facilitate model training, both sample types were manually annotated using the Labelme software (https://github.com/wkentaro/labelme), accessed on 6 June 2024, as shown in Figure 12. Carbon trace regions were labeled as 1, and non-trace regions as 0. Mask images aligned with each sample were generated to construct a comprehensive dataset for model training. Both the carbon trace images and their corresponding mask files were set to a resolution of 512 × 512 pixels.

## 5. Validation and Discussion of the Carbon Trace Segmentation Model

### 5.1. Training Platform Parameters and Evaluation Metrics

(1)Platform Parameters for Model Training

In order to validate the proposed model, a training environment has been constructed, and the detailed training parameters is shown in Table 1.

(2)Evaluation Metrics

To quantitatively assess the segmentation performance of the DSC-SeNet model, this study uses standard semantic segmentation metrics, including mean intersection over union (*M*_IoU_), pixel accuracy (*P*_A_), and precision (*P*_E_). *M*_IoU_ measures the average ratio of intersection to union between true and predicted pixel classes across all categories and serves as a comprehensive metric for segmentation accuracy. Pixel accuracy (*P*_A_) represents the ratio of correctly predicted pixels to the total number of pixels, while precision (*P*_E_) calculates the ratio of correctly identified positive sample pixels to the total number of positive samples. The formulas for these three metrics, derived from the confusion matrix, are given in Equations (17)–(19). Additionally, to evaluate the real-time segmentation capability, the inference speed is measured in frames per second (FPS).
(17)$$I_{\mathrm{mIoU}}=\left(\frac{T_{p}}{T_{p}+F_{p}+F_{n}}+\frac{T_{n}}{T_{n}+F_{n}+F_{p}}\right)\times 100\%$$
(18)$$P_{A}=\frac{T_{p}+T_{n}}{T_{p}+T_{n}+F_{p}+F_{n}}\times 100\%$$
(19)$$P_{E}=\frac{T_{p}}{T_{p}+F_{p}}\times 100\%$$
where *T*_p_ represents correctly identified true carbon trace pixels; *F*_p_ represents incorrectly identified true carbon trace pixels; *T*_n_ represents correctly identified carbon trace background pixels; and *F*_n_ represents incorrectly identified carbon trace background pixels.

### 5.2. Ablation Experiments

In this section, ablation experiments are performed to validate the effectiveness of each module proposed in the DSC-SeNet network. In the experiments, the STDC network [31] is used as a baseline for validation and analysis.

(1)Dual Threshold Setting in the Canny Operator

To enhance the visual features of carbon traces, the Canny operator was applied in the spatial path to generate supplementary spatial features. The upper and lower thresholds $\left(T_{\max},T_{\min}\right)$ are critical parameters of the Canny operator: pixels with gradient values above the upper threshold are classified as strong edges, those below the lower threshold are ignored, and those between the two thresholds are considered weak edges. Weak edges connected to strong edges are retained as true edges, while those unconnected are discarded as false edges. Thus, the choice of threshold values determines the detail richness in the resulting edge feature map. Figure 13 illustrates how different threshold settings $\left(T_{\max},T_{\min}\right)$ affect the detailed features of carbon traces. To identify the optimal threshold parameters, the DSC-SeNet was used as the segmentation network, keeping all other parameters unchanged, to analyze the impact of the threshold values on the mean intersection over union (*M*_IoU_). The edge feature maps, downsampled to 1/8 and 1/16 of the original image size, were used as supplementary spatial features for carbon traces and input to the SFRM module. Based on the results in Table 2, the final upper and lower thresholds were set to (40, 120).

(2)Validation and optimization of Canny edge detection

This section focuses on optimizing the selection of insertion layers and the size of Canny edge feature maps. Since each additional edge feature layer introduces more operations such as convolution and pooling, which will slow down inference speed, this analysis primarily considers cases where the number of edge feature layers is ≤2. Reducing the number and size of input layers results in a lower *M*_IoU_ but also limits the increase in computational complexity. Conversely, the improvement in segmentation performance tends to plateau as the number of layers and feature sizes increases. According to the experimental results in Table 3, considering both segmentation accuracy and inference speed, two feature maps at 1/8 and 1/16 of the original input size were finally selected as supplementary spatial features.

(3)Effectiveness of SFRM module

To improve the capability to perceive spatial features of carbon traces and to effectively integrate spatial paths with Canny edge features, the SFRM module is introduced in this study. To verify the effectiveness of the SFRM, activation visualization based on Grad-CAM was used to examine the module’s impact on spatial feature perception. As shown in Figure 14, without the SFRM, the segmentation model demonstrated limited perception of carbon trace edges, with significant detail loss that diminishes edge segmentation accuracy. In contrast, with the SFRM, the activation intensity around carbon trace edges showed a high gradient, with clear differentiation on either side of the edges, significantly improving boundary prediction and segmentation accuracy for carbon traces.

(4)Effectiveness of CAFA module

In order to effectively integrate the shallow spatial features and deep semantic features of carbon traces, the CAFA module was introduced in this study. As shown in Figure 15, when the CAFA module was not used, some carbon trace regions exhibited low activation intensity, resulting in incomplete activation and indicating insufficient utilization of semantic information. In addition, the activation gradient along the edges of the carbon trace was weak, resulting in suboptimal edge segmentation accuracy and suggesting limited use of spatial features. These observations revealed that without the CAFA module, the segmentation model struggled to effectively fuse semantic and spatial features, resulting in significant feature loss and reduced segmentation completeness and edge clarity. In contrast, the CAFA module facilitated more effective fusion of these two types of carbon trace features, significantly increasing the activation intensity and completeness of trace regions, which significantly improved both segmentation integrity and edge precision.

(5)Optimization of booster training strategy

To further improve segmentation accuracy, auxiliary segmentation heads were added at different stages along the semantic branch during training. Differences in segmentation performance at these stages were incorporated into the loss function to better guide model training, with the auxiliary heads removed during inference. To determine the optimal placement and number of segmentation heads, this section presents a comparative analysis of segmentation performance when heads were inserted at different stages. The results indicated that inserting segmentation heads at different points improved performance to varying degrees compared to training without auxiliary heads. Based on the accuracy data in Table 4, four segmentation heads were added from stage 2 to stage 5 as the final training strategy, with their placements highlighted in the gray areas of Figure 5.

(6)Quantitative comparison of different modules

To comprehensively evaluate the impact of the designed modules and training strategies on the segmentation performance, this section used the STDC network as a baseline and progressively incorporated the proposed modules and strategies. The training loss curves and mIoU curves on the validation set were analyzed. Since Canny edge features were to be integrated into the segmentation model through the SFRM module, the SFRM module was first introduced into the STDC network, followed by the Canny edge detection module and the CAFA module. As shown in Figure 16, during the training process, all three modules and the booster training strategy effectively reduced the training loss. For the validation set, these modules and strategies also contributed to improving segmentation performance. Table 5 summarizes the segmentation accuracy and inference speed on the test set. Among the modules, the Canny edge detection module yielded the most significant performance improvement, increasing mIoU by 3.3 percentage points, although it incurred the largest loss in inference speed. The SFRM module followed with a 1.6 percentage point increase in mIoU. The CAFA module further improved segmentation performance, improving mIoU by 0.9 percentage points. In addition, the booster training strategy provided a further 0.6 percentage point increase in mIoU without compromising inference speed, making it a highly efficient strategy for improving segmentation performance.

### 5.3. Comparison with State-of-the-Art Models

To validate the segmentation performance of the proposed model, this section presented a comparative analysis between DSC-SeNet and several state-of-the-art methods, including ICNet [26], Enet [27], PIDNet [28], SFNet [29], BiSeNet [30], STDC [31], BiSeNetV3 [32], SCTNet [33], HAFormer [34], BiSeNetV2 [43], CAS [44], GAS [45], TD4-PSP18 [46], HMSeg [47], and TinyHMSeg [47]. The experimental results showed that the proposed DSC-SeNet model achieved the best segmentation performance. As in Table 6, the bold number demonstrates that the corresponding model achieves the best performance in the corresponding metric. The segmentation accuracy and inference speed of ICNet and Enet were suboptimal, indicating that methods such as reducing sample resolution or using pruning to reduce network complexity, while simple and easy to implement, could not provide satisfactory results. By improving the efficiency of feature fusion, SFNet effectively improved the segmentation performance but still fell short of practical application requirements. BiSeNet, on the other hand, achieved significant breakthroughs in both segmentation accuracy and inference speed through innovative network design. Compared to ICNet, BiSeNet increased the mIoU by 19.3 percentage points and increased inference speed by more than two times. Based on BiSeNet, the STDC benchmark network further improved segmentation performance (mIoU) to 78.3% by optimizing the convolution modules in the feature extraction backbone, with only a slight trade-off in inference speed. Based on the STDC framework, the proposed DSC-SeNet model further improved the *M*_IoU_ of discharge carbon traces (mIoU) to 84.7%, a 6.4 percentage point improvement, by incorporating modules such as Canny edge detection, SFRM, and CAFA, and using the booster training strategy. As shown in the ablation study results in Section 4.2, the DSC-SeNet model performed exceptionally well in terms of both segmentation completeness and edge clarity, meeting the accuracy requirements for carbon trace extraction. However, due to the convolution and pooling operations introduced by the Canny edge detection and SFRM modules, the computational cost of the DSC-SeNet model was higher, and its inference speed was slower than that of the STDC model. Nevertheless, it still maintained an inference speed of over 90 FPS, which was sufficient to meet the real-time inspection requirements for transformer internal inspection robots.

To provide a more intuitive illustration of the segmentation performance of the proposed DSC-SeNet model, Figure 17 presents an inference speed vs. accuracy graph that compares DSC-SeNet to other state-of-the-art models. Models marked in blue represented segmentation models that used strategies such as lowering the input sample resolution and applying pruning techniques to reduce model complexity. These models often struggled to effectively balance segmentation accuracy with inference speed. Models marked in yellow represented the BiSeNet series, which used a dual-path framework for segmentation. This innovative structure reduced the loss of spatial detail and model complexity, resulting in significant improvements in both segmentation accuracy and inference speed. The STDC model, marked in orange, represented an enhancement of the BiSeNet framework by optimizing feature extraction. Furthermore, the proposed DSC-SeNet model improved shallow feature extraction, enhanced edge feature representation with Canny edge detection, and optimized spatial–semantic feature fusion along with booster training strategies. These enhancements resulted in a significant improvement in segmentation accuracy compared to state-of-the-art models.

### 5.4. Discussion

Achieving a balance between segmentation accuracy and inference speed is critical in the development of real-time semantic segmentation models. However, improving segmentation accuracy often comes at the cost of slower inference speed, as improvements in feature extraction and feature fusion typically require either additional modules or deeper network architectures, both of which reduce inference speed. To effectively manage this trade-off, it is essential to evaluate the dual impact of optimization strategies on both segmentation accuracy and inference speed. To better evaluate the effectiveness of these strategies, the ablation study used a ratio of inference speed reduction to segmentation accuracy improvement for each module, denoted as Δβ/Δα. This ratio reflects the reduction of inference speed per the increase in segmentation accuracy, with a smaller value indicating greater efficiency in performance improvement for the given module. As shown in Table 5, the Canny edge detection module had the highest Δβ/Δα ratio of 10.6, followed by the SFRM module with 7.3, the CAFA module with 4.0, and finally, the Booster Training strategy with 0.0.

These results demonstrated that although achieving the highest improvement in segmentation accuracy, the Canny edge detection module also resulted in a significant loss of inference speed, making it the least efficient in improving performance. In contrast, the performance gains of other modules and strategies were more moderate but more efficient in improving segmentation accuracy. In particular, the booster training strategy, applied only during model training, had no impact on inference speed, making it a highly cost-effective approach to improving segmentation accuracy. In future research, we will emphasize segmentation efficiency as a key evaluation criterion, focusing on the impact of improvement strategies on both computational complexity and segmentation accuracy. Specifically speaking, we intend to carry out network optimization on two aspects. On one hand, considering that the training-based methods have the advantage of increasing the dataset diversity and complexity, we plan to use training-based methods such as generative adversarial networks (GANs) and the segment anything model (SAM) to carry out data augmentation. On the other hand, inspired by SCTNet [33] and HAFormer [34], we plan to modify the transformer structure to reduce its computational requirement and incorporate it to construct more efficient and lightweight segmentation networks.

## 6. Conclusions

To address the need for accurate and rapid segmentation of discharge carbon traces in oil-immersed transformers, this paper proposed the DSC-SeNet, a lightweight model with a unilateral feature extraction framework. The DSC-SeNet enabled accurate and efficient segmentation of carbon traces characterized by diverse morphologies, large-scale variations, and irregular edges. This study provides technical support for assessing discharge severity and insulation status of oil-immersed transformers.

(1)Based on the STDC framework for feature extraction, DSC-SeNet incorporated a DFC module to improve adaptability to diverse morphologies of carbon traces. To streamline the network structure, a shallow feature-sharing mechanism was implemented to create a unilateral feature extraction network. To improve the ability to perceive and integrate both semantic and spatial features of carbon traces, a series of enhancement strategies—including Canny edge enhancement, SFRM, and CAFA—are employed. Therefore, the DSC-SeNet was constructed with these enhancements to meet the requirement for accurate and rapid segmentation of carbon trace.(2)Ablation experiments showed that the STDDC, SFRM, Canny edge, and CAFA modules, along with the booster training strategy, all contributed to improving segmentation performance. The Canny edge module provided the most significant improvement, increasing mIoU by 3.3 percentage points, although there was room to improve its efficiency. The CAFA module offered a smaller mIoU gain of 0.9 percentage points but achieved higher efficiency. The booster training strategy provided the smallest mIoU gain but had the unique advantage of being applied only during training, preserving inference speed.(3)With 512 × 512 carbon trace images as the input, DSC-SeNet outperformed state-of-the-art models in segmentation performance. Compared to the baseline STDC network, DSC-SeNet increased the mIoU from 78.3% to 84.7%, while achieving an inference speed of 94.3 FPS on an NVIDIA GTX 2050Ti. It indicated that the proposed model achieved a good balance between segmentation accuracy and inference speed, meeting the requirements for the micro-robot to perform accurate and real-time inspection in the transformers.

## Figures and Tables

**Figure 1 sensors-25-00043-f001:**
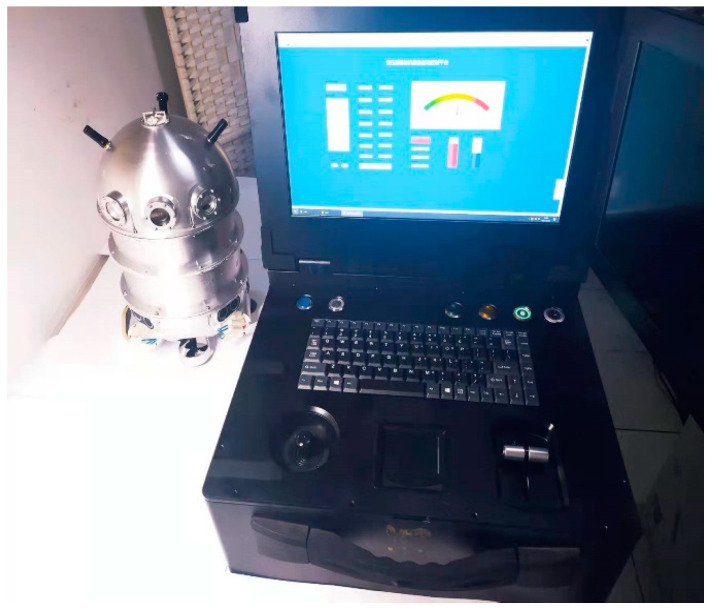
Physical drawing of the transformer internal inspection micro-robot.

**Figure 2 sensors-25-00043-f002:**
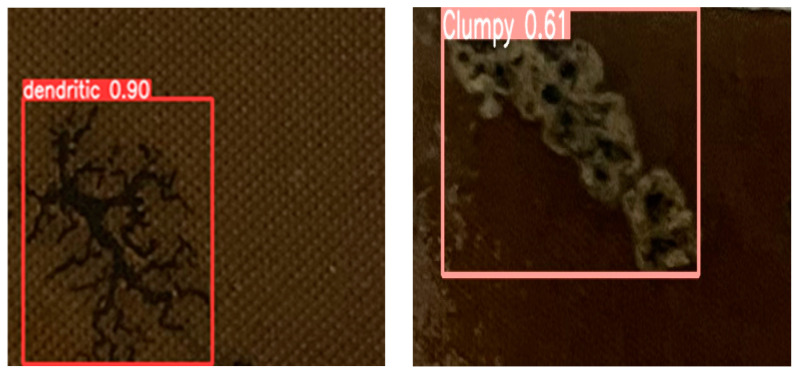
Discharge carbon trace identification based on improved YOLOv8 [17].

**Figure 3 sensors-25-00043-f003:**
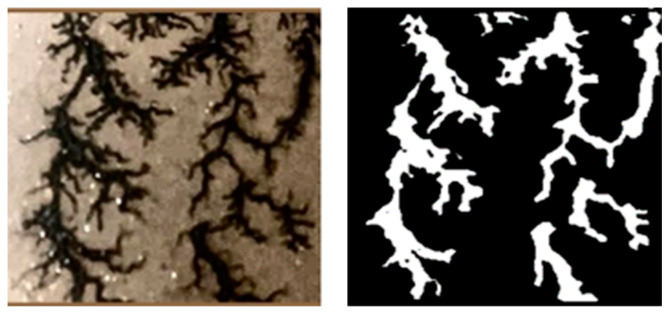
Segmentation of discharge carbon trace based on improved HCP-UNet [18].

**Figure 4 sensors-25-00043-f004:**
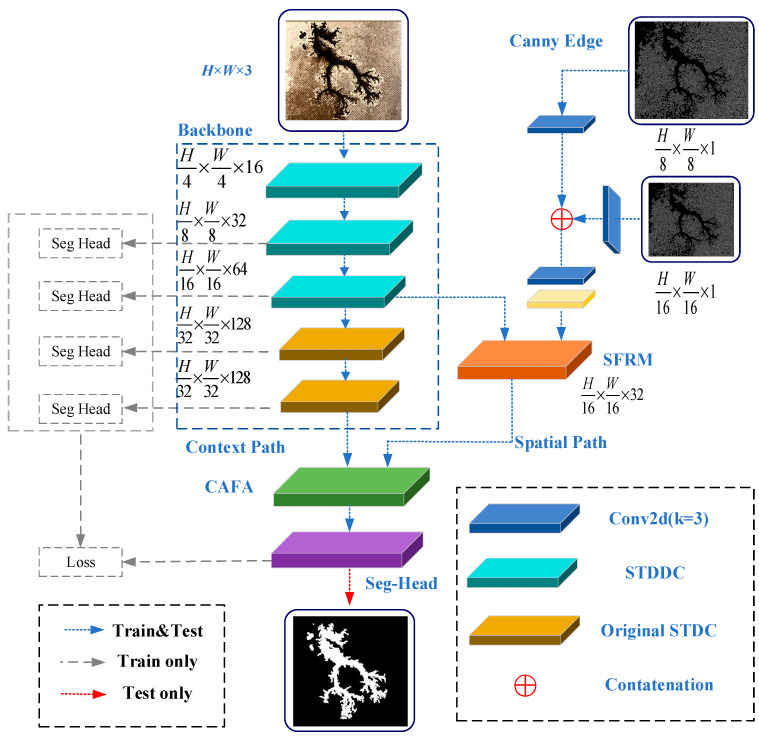
Network structure of the DSC-SeNet.

**Figure 5 sensors-25-00043-f005:**
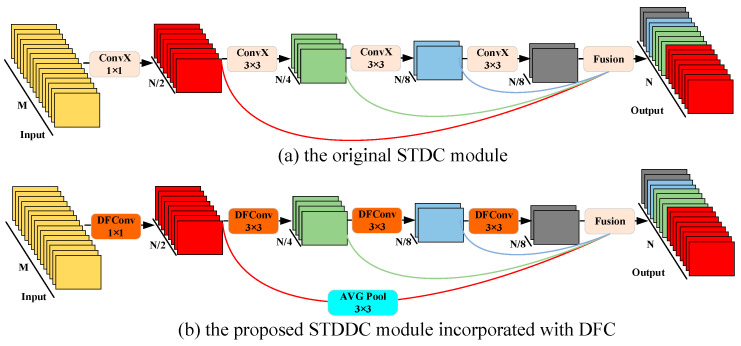
The STDC module and the STDDC incorporated with DFC.

**Figure 6 sensors-25-00043-f006:**
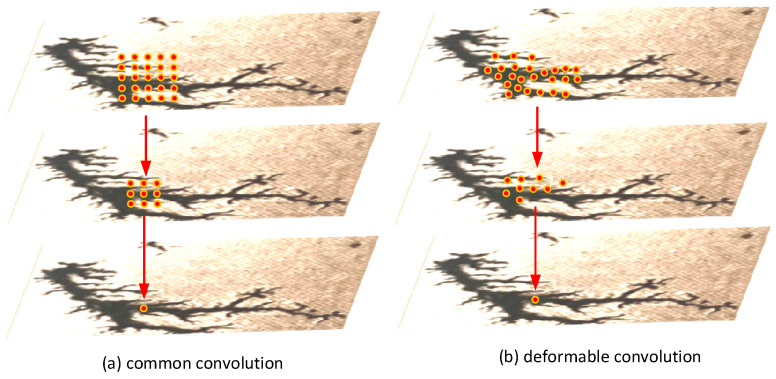
Receptive field differences between common and deformable convolutions.

**Figure 7 sensors-25-00043-f007:**
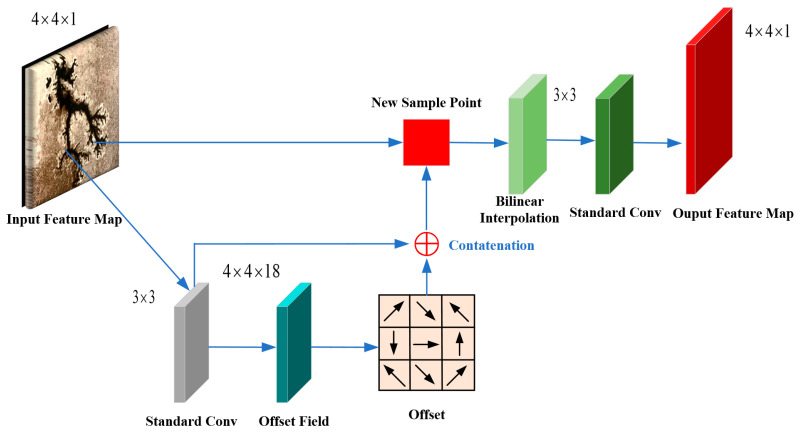
Network structure of the DFC.

**Figure 8 sensors-25-00043-f008:**
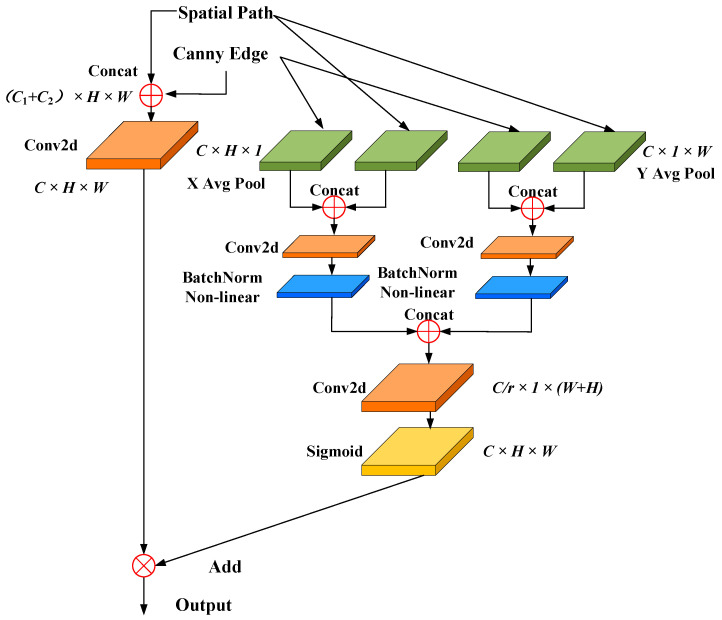
Network architecture of the SFRM.

**Figure 9 sensors-25-00043-f009:**
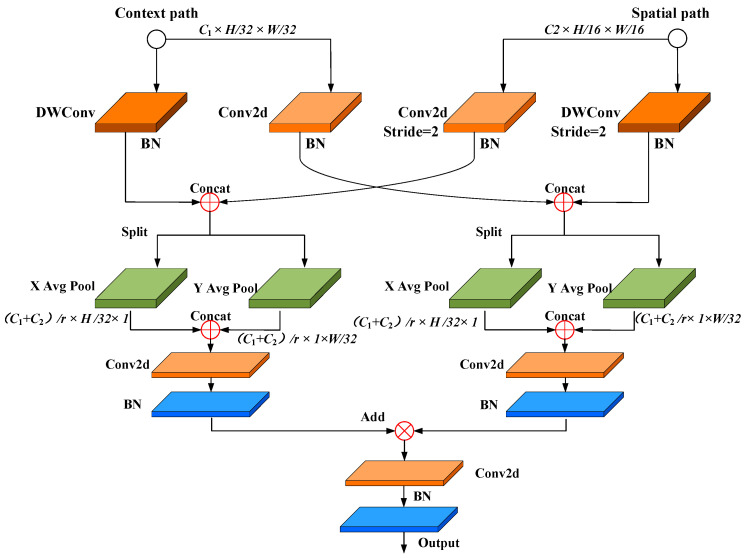
Network architecture of the CAFA.

**Figure 10 sensors-25-00043-f010:**
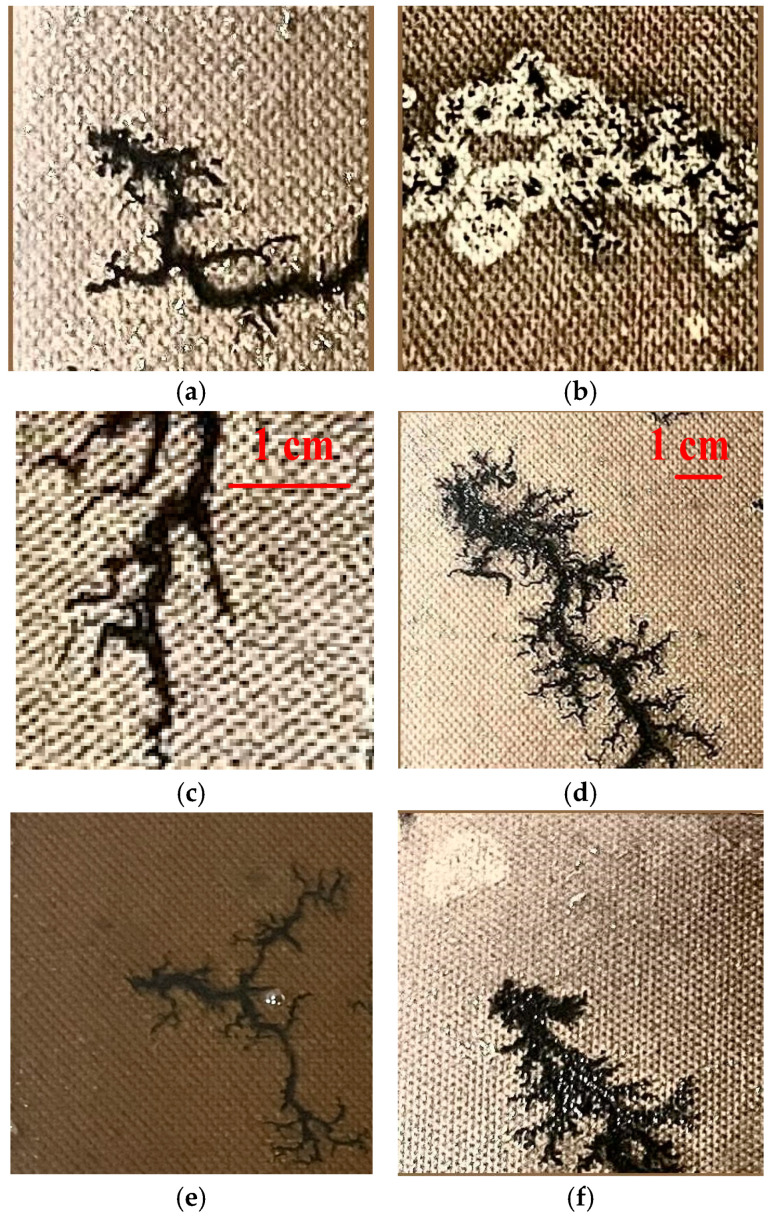
Examples of carbon trace samples. (**a**) dendritic carbon trace; (**b**) clustered carbon trace; (**c**) small-scale trace; (**d**) large-scale trace; (**e**) trace with low light; (**f**) trace with high light.

**Figure 11 sensors-25-00043-f011:**
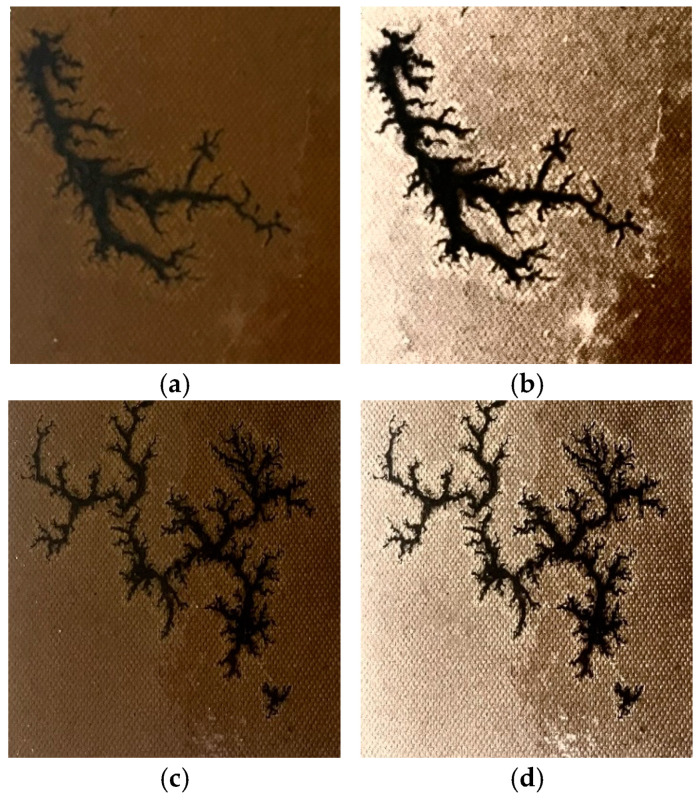
Comparison of carbon traces with and without improved MSRCR. (**a**) the original image#1; (**b**) the enhanced image corresponding to #1; (**c**) the original image#2; (**d**) the enhanced image corresponding to #2.

**Figure 12 sensors-25-00043-f012:**
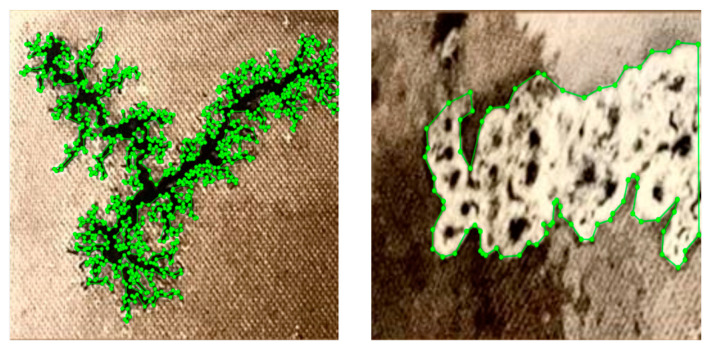
Labeling process of carbon trace with Labelme software.

**Figure 13 sensors-25-00043-f013:**
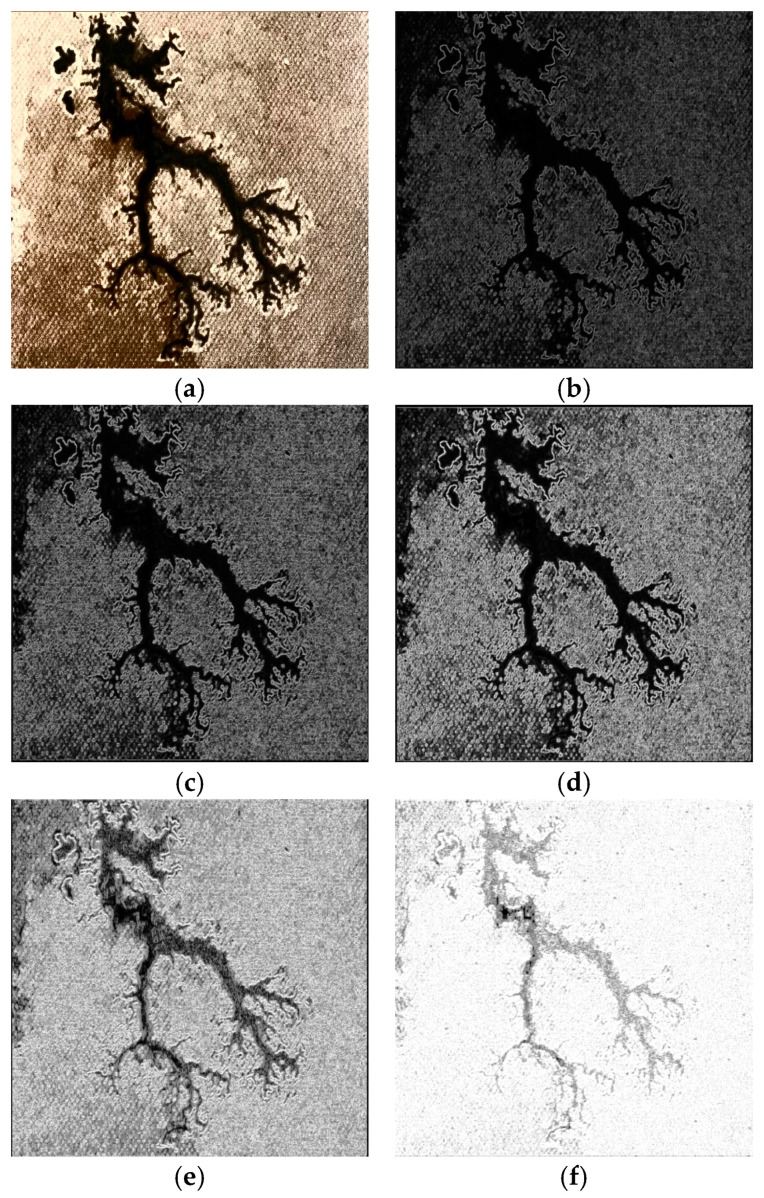
Visual comparison of Canny edge detection with different threshold values. (**a**) original; (**b**) (10, 30); (**c**) (25, 75); (**d**) (40,120); (**e**) (55, 165); (**f**) (70, 210).

**Figure 14 sensors-25-00043-f014:**
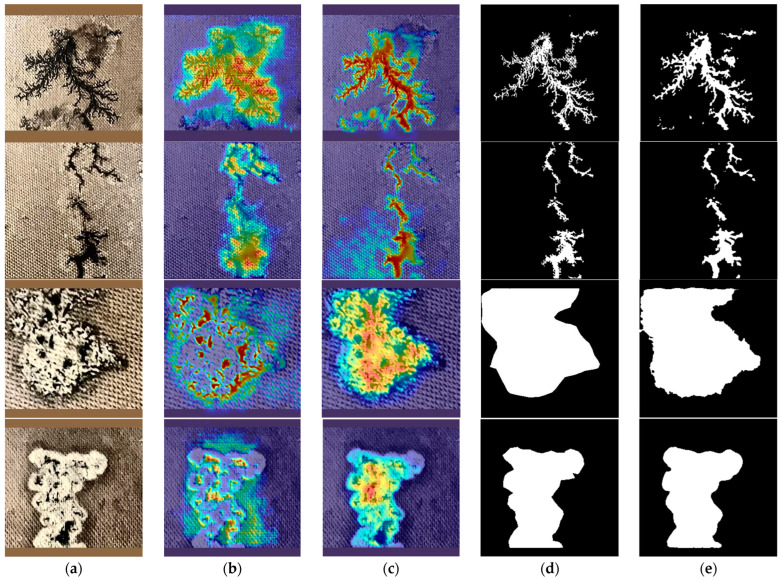
Grad-CAM comparison of segmentation results with and without the SFRM. (**a**) Input; (**b**) without SFRM; (**c**) with SFRM; (**d**) Groundtruth; (**e**) Prediction.

**Figure 15 sensors-25-00043-f015:**
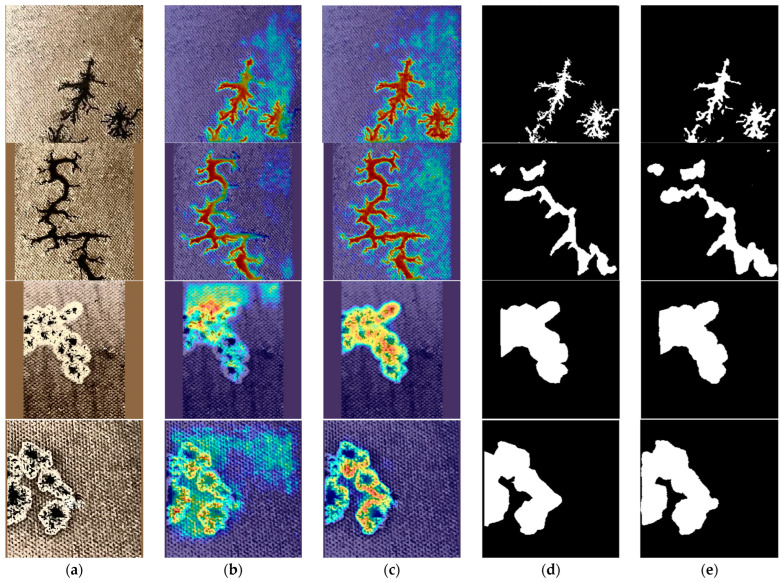
Grad-CAM comparison of segmentation results with and without the CAFA. (**a**) Input; (**b**) without CAFA; (**c**) with CAFA; (**d**) Groundtruth; (**e**) Prediction.

**Figure 16 sensors-25-00043-f016:**
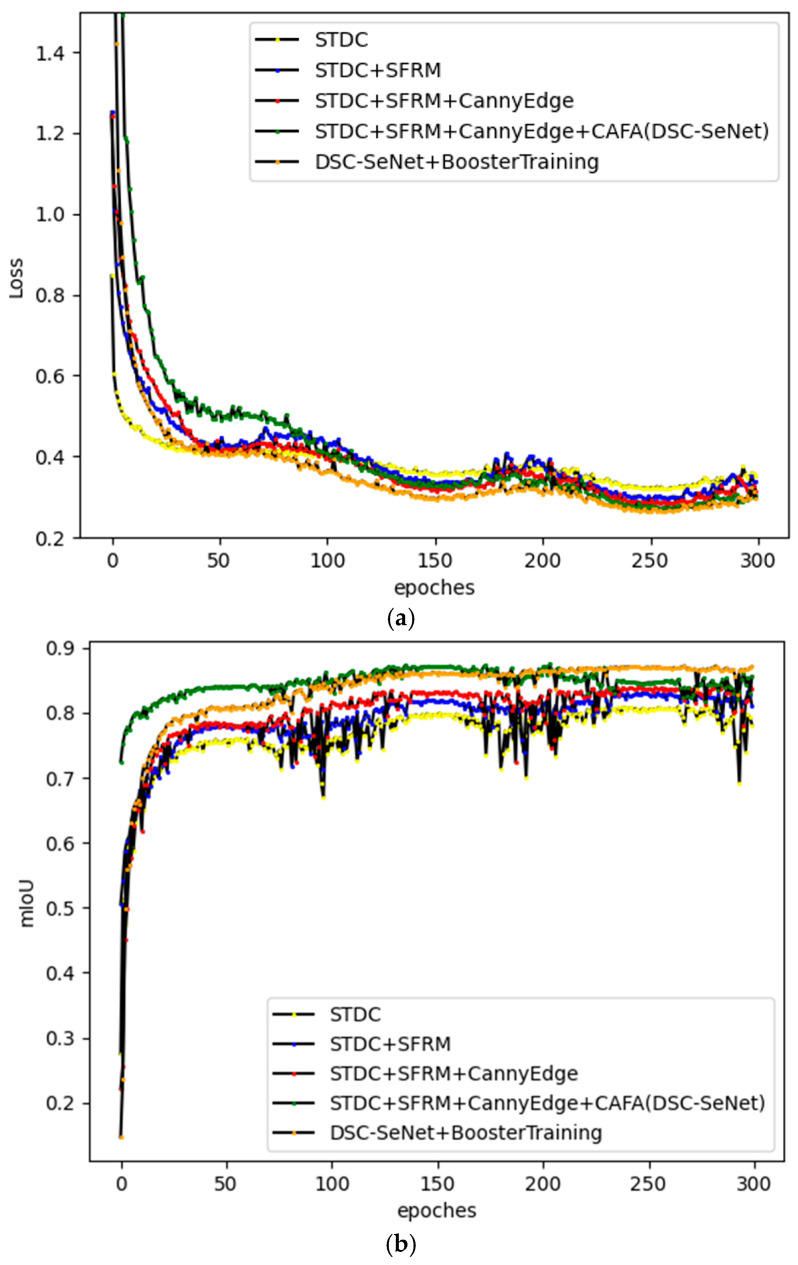
Performance comparison of segmentation model with different modules. (**a**) Loss curve; (**b**) mIoU curve.

**Figure 17 sensors-25-00043-f017:**
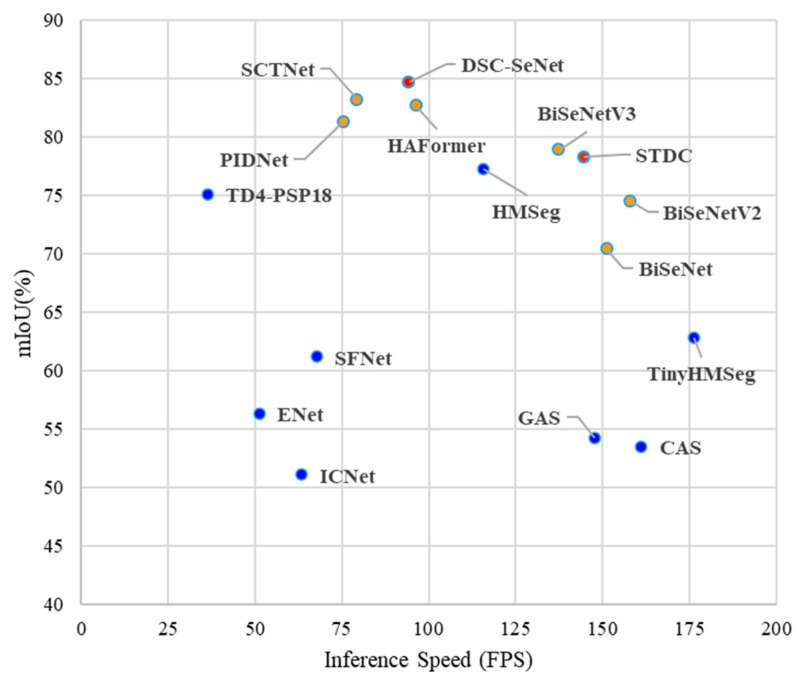
Comparison of DSC-SeNet and state-of-the-art models.

**Table 1 sensors-25-00043-t001:** Training environment and training parameters.

Training environment	Training platform	Windows 11 × 64
	CPU	Intel(R) Core(TM) i5-12500H@16 GB
	GPU	Nvidia GeForce RTX2050@4 GB
	Environment	Python 3.8.17
	Framework	PyTorch 2.0.1
	CUDA	11.8, CUDNN8.9.3
Training parameters	Optimizer	AdamW
	Initial learning rate	1 × 10^−3^
	Adjustment strategy for learning rate	CosineAnnealingLR
	Weight decay Coefficient	1 × 10^−2^
	Training epoches	300
	Batch size	8

**Table 2 sensors-25-00043-t002:** Quantitative comparison of Canny edge detection with different thresholds.

Feature Map Size	Threshold Value	mIoU(%)	FPS
1/8	(10, 30)	79.4	108.5
	(25, 75)	81.1	108.5
	(40, 120)	82.6	108.5
	(55, 165)	80.9	108.5
	(70, 210)	80.4	108.5
	(85, 255)	77.2	108.5
1/16	(10, 30)	78.8	131.6
	(25, 75)	80.4	131.6
	(40, 120)	81.7	131.6
	(55, 165)	81.5	131.6
	(70, 210)	80.8	131.6
	(85, 255)	80.5	131.6

**Table 3 sensors-25-00043-t003:** Comparison of Canny edge feature map numbers and sizes.

Map Num.	Edge Feature Size					mIoU(%)	FPS
	1/2	1/4	1/8	1/16	1/32		
1	**✓**					80.9	101.7
		**✓**				80.8	105.4
			**✓**			80.6	108.5
				**✓**		80.3	131.6
					**✓**	79.5	132.4
2	**✓**	**✓**				82.6	95.6
	**✓**		**✓**			82.4	96.4
	**✓**			**✓**		82.1	97.3
	**✓**				**✓**	81.9	98.5
		**✓**	**✓**			82.2	95.5
		**✓**		**✓**		82.6	96.2
		**✓**			**✓**	82.5	97.1
			**✓**	**✓**		83.2	97.9
			**✓**		**✓**	83.0	98.6
				**✓**	**✓**	82.3	128.4

**Table 4 sensors-25-00043-t004:** Comparison of booster position and number.

Stage1	Stage2	Stage3	Stage4	Stage5	mIoU(%)
					83.9
**✓**	**✓**				84.1
**✓**	**✓**	**✓**			84.5
**✓**	**✓**	**✓**	**✓**		84.2
**✓**	**✓**	**✓**	**✓**	**✓**	84.3
	**✓**	**✓**	**✓**	**✓**	84.7
		**✓**	**✓**	**✓**	84.4
			**✓**	**✓**	84.0

**Table 5 sensors-25-00043-t005:** Quantitative comparison of different modules.

Method	mIoU(%)	Δα↑	FPS	Δβ↓	Δβ/Δα
STDC network	78.3	-	144.6	-	-
STDC + SFRM	79.9	1.6	132.9	11.7	7.3
STDC + SFRM + CannyEdge	83.2	3.3	97.9	35.0	10.6
DSC-SeNet	84.1	0.9	94.3	3.6	4.0
DSC-SeNet + BoosterTraining	84.7	0.6	94.3	0	0.0

**Table 6 sensors-25-00043-t006:** Segmentation performance comparison of DSC-SeNet and state-of-the-art models.

Segmentation Method	mIoU(%)↑	PA(%)↑	PE(%)↑	FPS↑
ICNet	51.1	68.4	58.3	63.4
ENet	56.3	69.5	61.6	51.2
SFNet	61.2	70.3	65.9	67.7
BiSeNet	70.4	78.5	71.4	151.3
STDC	78.3	82.5	75.6	144.6
BiSeNetV3	78.9	84.3	77.1	137.5
BiSeNetV2	74.5	83.8	76.3	157.9
CAS	73.5	80.2	78.1	**161.2**
GAS	74.2	80.7	75.1	147.8
TD4-PSP18	75.1	84.3	79.2	36.3
HMSeg	77.3	85.6	79.4	115.7
TinyHMSeg	72.8	82.6	68.4	176.4
PIDNet	81.3	**89.3**	77.4	75.5
SCTNet	83.2	85.4	81.2	79.5
HAFormer	82.7	84.6	**82.1**	96.4
**DSC-SeNet**	**84.7**	87.3	81.4	94.3

## Data Availability

Data are contained within the article.

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
