# Peer review of "DSC-SeNet: Unilateral Network with Feature Enhancement and Aggregation for Real-Time Segmentation of Carbon Trace in the Oil-Immersed Transformer"

_sensors, 2024, doi:10.3390/s25010043_

Round 1
Reviewer 1 Report
Comments and Suggestions for Authors
The paper proposed a lightweight unilateral feature extraction framework DSC-SeNet, which was claimed to strike a balance between segmentation precision and inference speed, to address the challenge of real-time segmentation of carbon traces in oil-immersed transformers. The authors presented the network architecture and demonstrated its performance of image segmentation.
The topic in this paper is relevant and original within the field of transformer maintenance and inspection. The proposed research fills a significant gap of the traditional methods like dissolved gas analysis by introducing a novel segmentation model tailored for transformer inspection micro-robots.
This approach is helpful in addressing the challenges posed by the edges of carbon traces. Compared to existing works like ICNet, BiSeNet, and STDC, the proposed model produced a well-balanced improvement in accuracy (mIoU: 84.7%) and inference speed (94.3 FPS), making it more applicable for real-time deployment.
The paper organization and content is substantial for a publication. However, I have the following suggestions:
- Too many problems were claimed to be addressed, and the authors proposed a solution for each of them. How each part supports the final method? The authors should have a more clarified problem definition and its solution.
- The review of the previous work, to define the problem(s), was week. For example, data augmentation is important in image segmentation, how it affects your work? I suggest you to have a review on the following publications:
[1] Pre‐trained SAM as data augmentation for image segmentation, J Wu, Y Rao, S Zeng, B Zhang, CAAI Transactions on Intelligence Technology, https://doi.org/10.1049/cit2.12381
- Before give an abbreviation, the full name should be clarified and defined, e.g., the proposed method name DSC-SeNet.
- If possible, I advised the authors to expand the dataset to include more variations in transformer environments and carbon trace types to improve model generalizability.
- Can you give some comparisons to some transformer-based segmentation models like SAM?
- For Figure 4 (Network Structure), it is helpful to consider providing more detailed annotations or a legend to clarify module functionalities for readers less familiar with the model architecture
Reviewer 2 Report
Comments and Suggestions for Authors
The comments are attached in PDF.

Reviewer 3 Report
Comments and Suggestions for Authors
Large oil-immersed transformers are of crucial importance in modern power systems. However, their metal-enclosed housings make it difficult to directly conduct visual inspections on the internal insulation conditions. Traditional indirect monitoring techniques such as dissolved gas analysis often fail to accurately identify internal faults. Although visual inspection can quickly locate faults, it has problems such as long maintenance cycles, high risks to personnel and equipment, and high costs. With the development of robotics and artificial intelligence, miniature internal inspection robots have become a solution. For example, Txplore from ABB, SSTIR from Shenyang Ligong University, and the intelligent transformer inspection robot jointly developed by Tsinghua University and others can autonomously enter transformers for visual inspections, effectively overcoming the limitations of traditional methods. Regarding the key technology of visual recognition and analysis of internal discharge carbon traces in transformers,this paper proposes a lightweight unilateral feature extraction framework. It integrates deformable convolution (DFC), spatial feature refinement module (SFRM), coordinate attention feature aggregation (CAFA) and Canny edge enhancement to construct a real-time segmentation network, named DSC-SeNet. Experimental results show that the DSC-SeNet achieves a good balance between segmentation accuracy and inference speed. Taking an input of 512×512 as an example, it achieved 84.7% mIoU, which is 6.4 percentage points higher than that of the baseline STDC network, with a speed of 94.3 FPS on a NVIDIA GTX 2050Ti. This study provides technical supports for real-time segmentation of carbon traces and transformer insulation assessment. This is an interesting research paper. There are some suggestions for revision.
1. The motivation is not clear. Please specify the importance of the proposed solution.
2. The listed contributions are a little bit weak. Please highlight the novelty of the proposed solution.
3. The abstract of the article mentions the construction of a lightweight unilateral feature extraction framework, but the subsequent content does not provide an explanation. Moreover, the proposed network will be applied in actual miniature robots, yet details regarding practical aspects such as storage limitations are not described. Can you give some explanations in the paper?
4. Authors ignore some existing solutions, such as Glioma Segmentation-Oriented “Multi-Modal MR Image Fusion With Adversarial Learning, IEEE/CAA Journal of Automatica Sinica”, 9 (8), 1528-1531, and “Brain tumor segmentation based on the fusion of deep semantics and edge information in multimodal MRI”, Information Fusion 91, 376-387. Please discuss them in this paper.
5. The table names in many tables of the article are too close to the main text content, which is inconvenient for readers to read. Can you make some adjustments to the table layout?
6. In Figure 5, except for the annotations of the main proposed modules, the other different color blocks used in the backbone are not explained. Can you add annotations beside them for illustration?
7. In Table 6, this paper presents a comparison of the performance of different advanced models and that of the proposed model. However, there is a large amount of data. Can you highlight the data of the best performance for each item?
8. In the experimental part of the article, image enhancement was performed on the data based on the improved MSRCR algorithm. Was the subsequent comparative experiment also conducted on this basis? Can you emphasize this? The listed experimental results are not convincing. Please compare the proposed solution with more recently published solutions, especially the solutions published in 2024 and 2023.
Comments on the Quality of English LanguageNA
Round 2
Reviewer 3 Report
Comments and Suggestions for Authors
All my concerns have been addressed. I recommend this paper for publication.
Comments on the Quality of English LanguageNA